# Using Simplified Swarm Optimization on Multiloop Fuzzy PID Controller Tuning Design for Flow and Temperature Control System

**Ting-Yun Wu [1], Yun-Zhi Jiang [2],*, Yi-Zhu Su [3] and Wei-Chang Yeh [3]**

[1]   Department of Mathematics, National Tsing Hua University, Hsinchu 30013, Taiwan;
     s105021115@gapp.nthu.edu.tw
[2]   School of Mathematics and Systems Science, Guangdong Polytechnic Normal University,
     Guangzhou 510665, China
[3]   Integration and Collaboration Laboratory, Department of Industrial Engineering and Engineering
     Management, National Tsing Hua University, Hsinchu 30013, Taiwan; u102034007@gmail.com (Y.-Z.S.);
     yeh@ieee.org (W.-C.Y.)
*   Correspondence: jiangyunzhi@foxmail.com

**Abstract:** This study proposes the flow and temperature controllers of a cockpit environment control system (ECS) by implementing an optimal simplified swarm optimization (SSO) fuzzy proportional-integral-derivative (PID) control. The ECS model is considered as a multiple-input multiple-output (MIMO) and second-order dynamic system, which is interactive. In this work, we use five methods to design and compare the PID controllers in MATLAB and Simulink, including Ziegler–Nicolas PID tuning, particle swarm optimization (PSO) PID, SSO PID, and the combination of the fuzzy theory with PSO PID and SSO PID, respectively. The main contribution of this study is the pioneering implementation of SSO in a fuzzy PI/PID controller. Moreover, by adding the original gain parameters $K_p$, $K_i$, and $K_d$ in the PID controller with delta values, which are calculated by fuzzy logic designer, we can tune the parameters of PID controllers in real time. This makes our control system more accurate, adaptive, and robust.

**Keywords:** PID controller; parameter tuning; Ziegler-Nicolas; Particle Swarm Optimization (PSO); Simplified Swarm Optimization (SSO); fuzzy control

---

## 1. Introduction

It is important for an aircraft pilot to fly at a comfortable temperature and for the avionics in the cockpit to dissipate heat and operate smoothly. Therefore, in the cockpit environment control system (ECS), the temperature and air flow control are important technologies and issues. The system for temperature and air flow control is called the environmental control system. In recent years, the design of an ECS mostly involves mixing the hot air flow from the engine and the cold air flow cooled by the air conditioning package (i.e., the heat exchanger) to produce an appropriate air flow rate and temperature and transmit it into the cabin. The most critical technological issue for generating an appropriate air flow rate and temperature is to control the cold and hot air flows and mix them to reach the desired value. It is a complex multiple-input multiple-output (MIMO) problem [1].

In closed loop control systems, the proportional-integral-derivative (PID) controller design has been widely used in the industry [2]. Its advantages include good stability, easy adjustment, and high reliability. The history of PID control can be traced back to the governor design in the 1890s [3,4]. Subsequently, it was gradually developed for ship automatic operating systems, advanced aircraft flight controllers, and related engine controllers. Currently, PID control is widely used in electronic

systems for positioning of magnetic disk read–write heads [5], spacecraft attitude control [6], and power systems [7].

Obviously, we also apply PID control as the design principle of the controller in this study. PID control is defined as the sum of the following three terms: the error term multiplied by the proportional gain parameter $K_p$, the error integral term multiplied by the integral gain parameter $K_i$, and the error derivative term multiplied by the derivative gain parameter $K_d$. How to obtain the optimal $K_p$, $K_i$, and $K_d$ for PID control is an important issue to be discussed and explored [8–10]. However, in most studies, only a local optimal solution is reached or these studies might be constrained by time-varying, nonlinear, and MIMO systems rather than by a single-input single-output (SISO) system, which can be easily manipulated [11,12]. Therefore, this study aims to improve the traditional methods and use optimal heuristic algorithms to compare and analyze, and then simulate and evaluate, the ECS of an aircraft cockpit to improve the air flow and temperature control.

There is still room for improvement in PID control; thus, we attempt to optimize the PID controller with particle swarm optimization (PSO) or simplified swarm optimization (SSO) algorithms. The PSO algorithm is a metaheuristic algorithm [13]. It is a model that simulates the flight status of a flock of birds in search of food [14]. According to the adaptability of a flock of birds to the environment, an individual bird can drive the group to move toward the best position with the most adequate food. PSO is often used to solve optimization problems because it requires few or even no assumptions [15,16]. It can also provide multiple possible solutions. This study will use PSO to find the three gain parameters $K_p$, $K_i$, and $K_d$ of the PID controller because of its good optimization effect and simple operations.

SSO was proposed by Yeh in 2008 [17]. It is an improved algorithm to resolve the shortcomings of PSO, which only reaches a local optimal solution instead of the global optimal solution. The general concept and purpose of SSO are the same as those of PSO, but SSO uses more random variables to change the optimization method and optimize the results. SSO allows the optimization result to be not only limited to the local optimum, but also to achieve a global optimum. Thus, this study utilizes SSO to find the gain parameters of the PID controller and compares it with the optimization result of PSO.

To describe the problems for the ECS of an aircraft cockpit, we also apply the fuzzy theory and combine it with PSO or SSO. The fuzzy theory is widely applied in the fields of engineering, humanities, and society [18]. The fuzzy theory is used to replace the binary logic concept with multiple fuzzy sets, and it can also use membership functions to classify situations expressed by exact values into a suitable fuzzy set [19]. This study utilizes the fuzzy logic designer in MATLAB to realize one type of fuzzy control, called the Mamdani-type fuzzy controller [20]. We combine PSO and SSO with the fuzzy theory and compare their experimental results. We control the system by adding three initial values, $K_p$, $K_i$, and $K_d$, calculated by PSO and SSO, respectively, to time-varying delta values (i.e., tuned values) from the fuzzy theory.

The remainder of this paper is organized as follows. Section 2 introduces the mathematical model of the aircraft cockpit ECS and the theoretical background needed for the controller design (i.e., PID Control), PSO, SSO, and fuzzy theory. Section 3 proposes the design and simulation methods of the controller for the aircraft cockpit ECS. Then, we present the results in Section 4. Finally, we conclude with the contributions of this study in Section 5.

## 2. Methodological Background

This section introduces the related methodological background, including the mathematical model of the aircraft cockpit ECS, PID Control, PSO, SSO, and fuzzy theory.

### 2.1. Mathematical Model of Airplane Cockpit ECS

In the ECS of an aircraft cockpit, the controlled system uses two valves, called butterfly valves, to control the amount of cold and hot air flows. The opening and closing degrees of the butterfly valve flaps, which affect the cross-sectional area of the tube, are controlled by a torque motor driven by electric current. The change in cross-sectional area of the tube impacts the air flow passing through.

The cold and hot air flows are mixed according to the desired air flow temperature and flow rate that the pilot wants to achieve, and then they are sent to the cockpit. Temperature and flow sensors are installed to detect the mixed air flow and facilitate the feedback signals to the system controller to calculate the amount of errors for the air flow temperature and flow rate, thus, forming a closed loop control system. In such a closed loop system, according to the feedback of errors, the PID controller regulates its gain parameters $K_p$, $K_i$, and $K_d$ to be more suitable for the air flow temperature and flow rate required by the pilot. This closed loop system for the ECS of an aircraft cockpit is shown in Figure 1 [1].

The system in Figure 1 is modeled as shown in Figure 2. We use $C_1$ to represent the flow controller and $C_2$ to represent the temperature controller. After the errors pass through the controller, currents $i_1$ and $i_2$ are obtained and then sent to the controlled system. The two butterfly valves, $V_h$ and $V_c$, and the controlled system can be expressed by four transfer functions as the following mathematical formula by the Laplace transform:

$$G_p(s) = \begin{bmatrix} G_{11}(s) & G_{12}(s) \\ G_{21}(s) & G_{22}(s) \end{bmatrix} = \begin{bmatrix} G_{11}(s) & -G_{11}(s)K_c \\ G_{21}(s) & G_{21}(s)K_h \end{bmatrix} \tag{1}$$

$$= \begin{bmatrix} \frac{-1.908s+52.2481}{s^2+34.48s+52.2481}*0.2433 & \frac{-1.908s+52.2481}{s^2+34.48s+52.2481}*(-2.311) \\ \frac{-6.917s+161.9432}{s^2+44.87s+161.9432}*0.3023 & \frac{-6.917s+161.9432}{s^2+44.87s+161.9432}*12.24 \end{bmatrix} \tag{2}$$

$G_{11}$:  the transfer function with which the $V_c$ valve flap (cold air source) input current led to a change in the mixed air flow.

$G_{12}$:  the transfer function with which the $V_c$ valve flap (cold air source) input current led to a change in the mixed air temperature.

$G_{21}$:  the transfer function with which the $V_h$ valve flap (hot air source) input current led to a change in the mixed air flow.

$G_{22}$   the transfer function with which the $V_h$ valve flap (hot air source) input current led to a change in the mixed air temperature.

$K_h$:  gain value of the heat flow to the temperature change of the mixed gas.

$K_c$:  gain value of the cold flow rate to the temperature change of the mixed gas.

The constant values, which are actually tested by experimental methods in Equation (2) are suggested according to reference. $G_p(s)$ represents the output air flow and temperature via the controlled system when the inputs are currents $i_1$ and $i_2$.

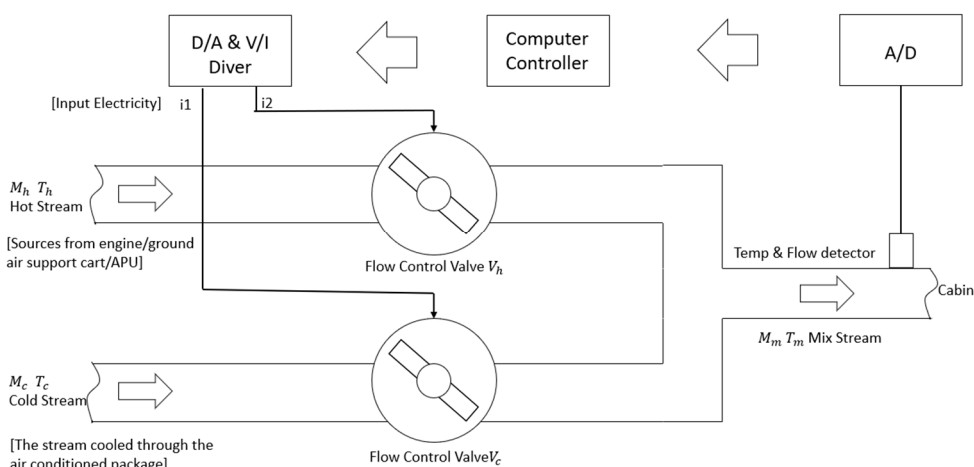

**Figure 1.** Diagram of cockpit flow and temperature control system.

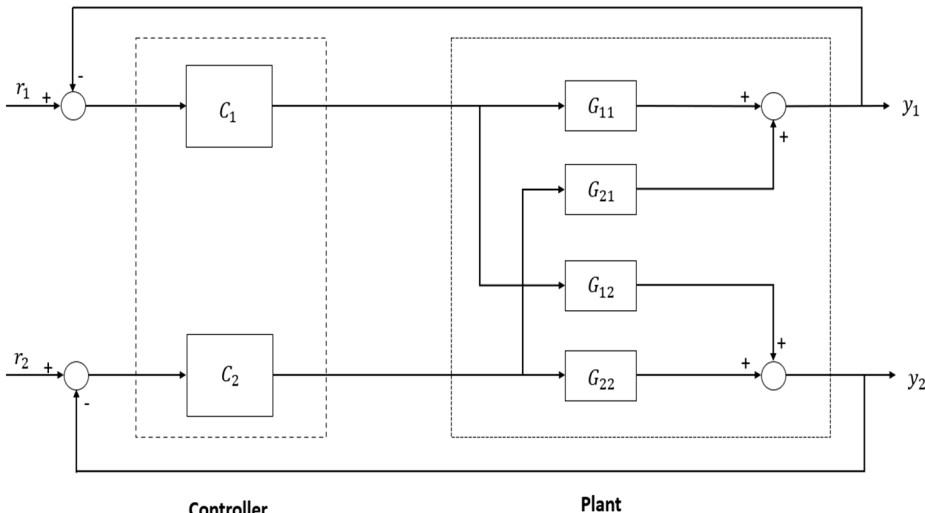

**Figure 2.** Flowchart modeling the environmental control system.

## 2.2. PID Control

The PID controller design is used in this study to control the air temperature and flow. A PID control consists of three units: proportional ($K_p$), integral ($K_i$), and derivative ($K_d$). The so-called control tunes the gain values of these three units. $u(t)$ is defined as the output function of the controller at time $t$, and then a PID controller can be expressed as the following mathematical formula [2]:

$$u(t) = K_p E(t) + K_i \int_0^t E(\tau)d\tau + K_d \frac{d}{dt}E(t) \tag{3}$$

$$u(s) = (K_p + \frac{K_i}{s} + K_d s)E(s) \tag{4}$$

where

$K_p$: coefficient of the proportional term.
$K_i$: coefficient of the integral term.
$K_d$: coefficient of the derivative term.
$E$: error = set value − feedback value
$t$: current time.
$\tau$: integral variable, with values from 0 to $t$.
Equation (4) is an expression defined in the continuous s-domain.

### 2.2.1. Effects of $K_p$, $K_i$, and $K_d$ on the PID Controller

The coefficient of the proportional term $K_p$ is directly multiplied by the error term $E$ so the input error term $E$ is proportional to the output $K_p*E$. If only $K_p$ is used in the controlled system and it is a non-first-order system (type one), there will be a steady-state error in the output [21]. Generally, steady-state errors will be eliminated by $K_i$ to achieve system stability, which will be explained in the next paragraph.

To eliminate the steady-state error, the integral control term $K_i$ can be added to increase the bias term in the system (i.e., biasing). Integrating the error term over a period of time and multiplying it by the integral gain value $K_i$ yields the output value of the integral control. Over time, even if the error is small, the value of the error integration will become increasingly larger, causing the steady-state error to gradually decrease until it equals zero.

The differential control term $K_d$ utilizes the first-order differentiation to consider future errors. It is multiplied with the first derivative of the error as the output value. For example, when the automatic controlled system adjusts the error, the system may oscillate or even be unstable. At this time, the

differential control can predict the trend of the error change. The larger the value of differentiation is, the more rapidly the controlled system can react.

The three gain control parameters in PID control do not necessarily have to be used at the same time. Instead, they can be selected according to different application requirements to reduce the computational burden. For example, a proportional and derivative controller (PD controller) can improve the dynamic characteristics of the system during the adjustment process. However, although the rise time of the PD controller is shorter, it cannot effectively suppress the steady-state errors of the system. A proportional and integral controller (PI controller) has a slower response time. However, as it can eliminate the steady-state error, it is often applied in the industry. In this paper, we supposed to use "PID controller" just for generic name, but may including PI controller or PD controller.

### 2.2.2. Adjustment of PID Gain Parameters

An excellent PID controller also needs to be matched with suitable PID gain parameters. Although it seems easy to tune these three variables, they correspond to different PID gain parameters for different applications and system transfer functions, and may even be in conflict with each other. Thus, the PID gain parameters require tradeoffs to achieve the best overall control results. There are many methods to find suitable PID gain parameters under this type of control problem, for example, the Ziegler–Nichols [8] and Cohen–Coon [9] tests.

The most famous PID gain parameter adjustment method is the Ziegler–Nichols test, which belongs to the empirical formula proposed by Ziegler and Nichols in 1912. The PID controller adjusted by the Ziegler–Nichols test is called the Z–N tuning method. The Z–N tuning method is proposed for a general system and a first-order transfer function model with delay. The Z–N tuning method is one of our strategies for modifying the PID controller. Its model can be expressed as

$$G(s) = \frac{Ke^{-Ls}}{Ts + 1} \tag{5}$$

### 2.3. Particle Swarm Optimization (PSO)

PSO is an evolutionary computing technology proposed by J. Kennedy and R. C. Eberhart in 1995 [14], which is suitable for solving non-linear problems. The characteristic of PSO is that it simulates the flight behavior of birds (swarms) to find the place with the most adequate food. An individual bird (i.e., a particle) considers its current position, its best position in the past, and the best position of the group from before to now, and then it weighs the three terms to find its next flight position. By calculating the fitness value of each position through iteration and the fitness function of the PSO algorithm, we can judge the pros and cons of each bird's (or particle's) current position and move toward the direction of optimization by the updated mechanism.

### 2.3.1. Algorithm of PSO

Assume that the *i*th particle among the group can be represented as $X_i = (x_{i1}, x_{i2}, x_{i3}, \ldots, x_{id})$ in a *d*-dimension space. The best previous positions of the *i*th particle can be represented as pbest = ($pbest_{i1}$, $pbest_{i2}$, $pbest_{i3}$, $\ldots$, $pbest_{id}$). The best particle in the group can be represented as gbest. The velocity of the *i*th particle can be represented as $V_i = (v_{i1}, v_{i2}, v_{i3}, \ldots, v_{id})$. The updated formula for the particle velocity and particle position can be expressed as $v_{im}{}^{t+1}$ and $X_{im}{}^{t+1}$, respectively.

$$v_{im}{}^{t+1} = w^*v_{im}{}^t + C_1{}^*rand^*(pbest_{im} - X_{im}{}^t) + C_2{}^*rand^*(gbest_{im} - X_{im}{}^t) \tag{6}$$

$$X_{im}{}^{t+1} = X_{im}{}^t + v_{im}{}^{t+1} \text{ for } i = 1, 2, 3, \ldots, n \text{ and } m = 1, 2, 3, \ldots, d \tag{7}$$

where

　　*n*: number of particles.
　　*d*: dimension.

$t$: number of iterations.
$v_{im}{}^t$: velocity of particle at iteration $t$.
$w$: inertia weight factor.
$C_1, C_2$: acceleration constant.
*rand*: random number between 0 and 1.
$X_{im}{}^t$: current position of $i$th particle at iteration.
pbest: best previous position of the $i$th particle.
gbest: the best particle among all the particles in the swarming population.
The flowchart of PSO is shown in Figure 3.

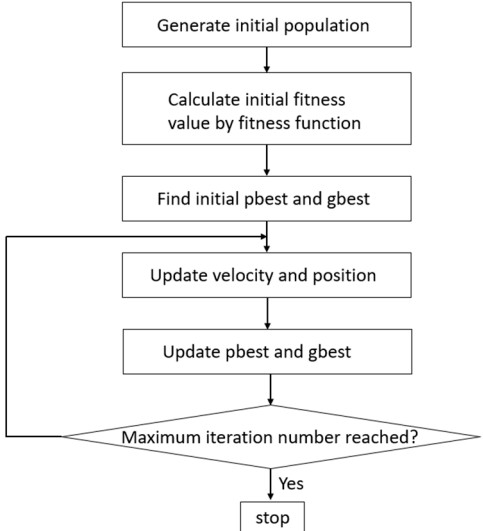

**Figure 3.** Flowchart of particle swarm optimization (PSO).

### 2.3.2. Fitness Function of PSO

In different applications, the suitable fitness functions are different, depending on the optimization problem defined by the corresponding application. For example, in the shortest path problem, the smallest sum of the distances from a location to all destinations is the best solution. Thus, the fitness function of the shortest path problem is designed to calculate the total distance of a location from all destinations. The smaller the output value of the fitness function, the closer the location is to the optimal location.

However, PSO suffers from premature convergence, especially in high-dimensional multimodal problems. The convergence speed decreases as the number of iterations increases. These facts lead to difficulties for the particles to achieve the best global fitness values. Therefore, in the next section, we introduce SSO.

### 2.4. Simplified Swarm Optimization (SSO)

SSO is a modified algorithm proposed to overcome the disadvantages of PSO [22]. The advantages of SSO are its simple, effective, and flexible characteristics [23–29]. The SSO algorithm is similar to the PSO algorithm. SSO also applies the concepts of pbest and gbest, and introduces the concept of randomness to update the location to ensure population diversity and enhance the ability of the algorithm to find the global optimum and avoid shackling in local optimums. It is worth noting that the main contribution of this study is the pioneering implementation of SSO in a fuzzy PID controller.

### 2.4.1. Algorithm of SSO

**STEP S0**. Generate $X_{0i}$, calculate $F(X_{0i})$, let $X_{0i} = P_i$ and $t = 1$, where $I = 1, 2, \ldots, N_{pop}$.
**STEP S1**. Let $I = 1$.

**STEP S2**. Update $X_{t-1i}$ to $X_{ti}$, based on Equation (8) and calculate $F(X_{ti})$:

$$X_{ij}^{t+1} = \begin{cases} X_{ij}^t & \text{if } \rho \epsilon [0, \ c_w] \\ p_{ij} & \text{if } \rho \epsilon [c_w, c_p] \\ p_{gbestj} & \text{if } \rho \epsilon [c_p, c_g) \\ X & \text{if } \rho \epsilon [c_g, \ 1) \end{cases} \quad (8)$$

**STEP S3**. If $F(X_{ti})$ is better than $F(P_i)$, then let $P_i = X_{ti}$. Otherwise, go to STEP S5.

**STEP S4**. If $F(P_i)$ is better than $F(P_{gbest})$, then let gbest = $i$.

**STEP S5**. If $I < N_{pop}$, let $I = I + 1$ and go to STEP S2.

**STEP S6.** If $t = N_{gen}$ and/or CPU time are met, then halt; otherwise, *let $t = t + 1$ and go back to STEP S1.*

where

$X_{ti}$: $X_{ti} = (X_{ti1}, X_{ti2}, \dots , X_{tiNvar})$ is the $i$th solution at the $i$th generation, where $I = 1, 2, \dots , N_{pop}$ and $t = 1, 2, \dots , N_{gen}$.

$F()$: the fitness function value of the solution.

$P_i$: $P_i = (p_{i1}, p_{i2}, \dots , p_{iNvar})$ is the current pbest with respect to the $i$th solution, where $I = 1, 2, \dots , N_{pop}$.

$N_{pop}$: the number of SSO solutions in each generation.

$\rho$: random number uniformly distributed in [0, 1].

$C_w, c_p, c_g$: parameters that represent the probabilities of the new variable value generated from the current solution, pbest, and gbest.

Pbest, gbest: pbest represents the best individual solution achieved so far; gbest represents the best value of all solutions so far.

$N_{gen}$: total number of independent generations in SSO.

### 2.4.2. Fitness Function of SSO

In SSO, the fitness function is still the same as that of PSO. It is used to determine the pros and cons of a position, whether it is getting closer to the optimization. Moreover, the output value of the fitness function represents the optimization degree of the position [22].

### *2.5. Fuzzy Theory*

The concept of fuzzy sets was first proposed by Professor L.A. Zadeh of the University of California, Berkeley in 1965 [30]. Fuzzy logic was used to describe the nature of things in reality instead of the binary logic description used in the past. The binary logic only describes the characteristic of a set by values of 0 and 1, whereas the fuzzy set utilizes infinitely numerous points of membership function values to describe a set. Fuzzy logic effectively improves the shortcomings of binary logic descriptions.

### 2.5.1. Mamdani Fuzzy Inference Method

Professor E.H. Mamdani at the University of London and his research group designed in 1974 a controller composed of fuzzy parameters and fuzzy rules and applied it to the control of boilers and steam engines. It successfully simulated human reasoning and obtained good control results [20]. The fuzzy logic control directly adopts linguistic control rules, and thus it is very suitable for systems where mathematical models are difficult to obtain, their dynamic characteristics are difficult to handle, or the situation changes are very significant.

The general fuzzy control system basically includes four parts: fuzzification, fuzzy rule base, fuzzy inference mechanism, and defuzzification. In general, the monitor in the system measures a concrete value, but to match the verbalized conditional rules of the fuzzy logic controller, the variable ranges of the input and output variables are divided into several fuzzy sets. Hence, the value is first fuzzified and appointed to the fuzzy set where it belongs. The fuzzy rule base is the judgment

criterion of the entire controlled system, which is assembled by experience and human wisdom. It is expressed by the conditional expression of "IF-THEN" and various possible situations of the controlled system are described in logical sentences. The fuzzy inference mechanism is based on the theory set by the fuzzy rule base and obtains a fuzzified output. The final step is to defuzzify the output of the fuzzification to transfer an exact value to the plant. A small example is used to illustrate the procedure of the Mamdani fuzzy inference method [31–33].

### 2.5.2. Principles of Fuzzy PID Controller Design

We aim to improve the traditional PID controller by modifying the input error E and error change (differential) ΔE, by adding the original gain parameters $K_p$, $K_i$, and $K_d$ with delta values [12]. It has been proven by many researches that fuzzy PID controllers are designed in the popular and effective way by adding the original gain parameters $K_p$, $K_i$, and $K_d$ in PID controller with delta values. According to the definition of adaptive in control area, it is clear that our controller is adaptive, since it can tune the gain parameters according to different errors and the derivative of errors over time in our controller. Therefore, the improved PID controller has favorable dynamic characteristics and revises over time to allow the system to achieve better control effects. The mathematical expressions are as follows:

$$K_p = K_p' + \Delta K_p \tag{9}$$

$$K_i = K_i' + \Delta K_i \tag{10}$$

$$K_d = K_d' + \Delta K_d \tag{11}$$

where $K_p'$, $K_i'$, and $K_d'$ are the initial values of the PID controller; $\Delta K_p$, $\Delta K_i$, and $\Delta K_d$ are the outputs of delta value from the fuzzy PID controller; $K_p$, $K_i$, and $K_d$ are the final PID gain parameters of the controlled system.

## 3. Proposed Strategies

In this study, five strategies are proposed to design the controller of the aircraft cockpit ECS, namely, the Ziegler–Nichols test (Z–N), PSO, SSO, fuzzy theory combined with PSO, and fuzzy theory combined with SSO.

### 3.1. Ziegler–Nichols Tuning

We use the relay feedback method [34] to obtain the parameters $K_u$ and $T_u$. Then, we utilize the Z–N tuning method to obtain and set the PID gain parameters $K_p$, $K_i$, and $K_d$, as presented in Table 1. The steps of the relay feedback method are as follows:

**Table 1.** Ziegler–Nichols test (Z–N) tuning method.

| Controller | $K_p$ | $K_i$ | $K_d$ |
|---|---|---|---|
| P | $0.5\,K_u$ | NA | NA |
| PI | $0.45\,K_u$ | $0.83\,T_u$ | NA |
| PID | $0.6\,K_u$ | $0.5\,T_u$ | $0.125\,T_u$ |

**STEP S1.** In the original system, the PID controller is changed to the relay block, as shown in Figure 4.

**STEP S2.** The on–off control of the relay block to lead the system generates delay and disturbance, where the on–off control is simulated by +1 and −1, as shown in Figure 5.

**STEP S3.** The oscillation curve of the system is shown in Figures 6 and 7.

**STEP S4.** $T_u$ and $a$ are obtained, as shown in Figures 6 and 7. $T_u$ and $a$ are substituted into the formula $K_u = \frac{4}{\pi} \times \frac{d}{a}$ to calculate $K_u$, where $T_u$ is the oscillation period, $a$ is the amplitude, $K_u$ is the critical gain value, and $d$ is the output value of Relay (which has been set to 1).

From the above four steps, and from Figures 6 and 7, we can obtain $T_u$ and $K_u$ of the PID flow controller and PID temperature controller as follows:

PID flow controller: $T_u = 0.24$, $K_u = 3.4571$.

PID temperature controller: $T_u = 0.24$, $K_u = 0.3596$.

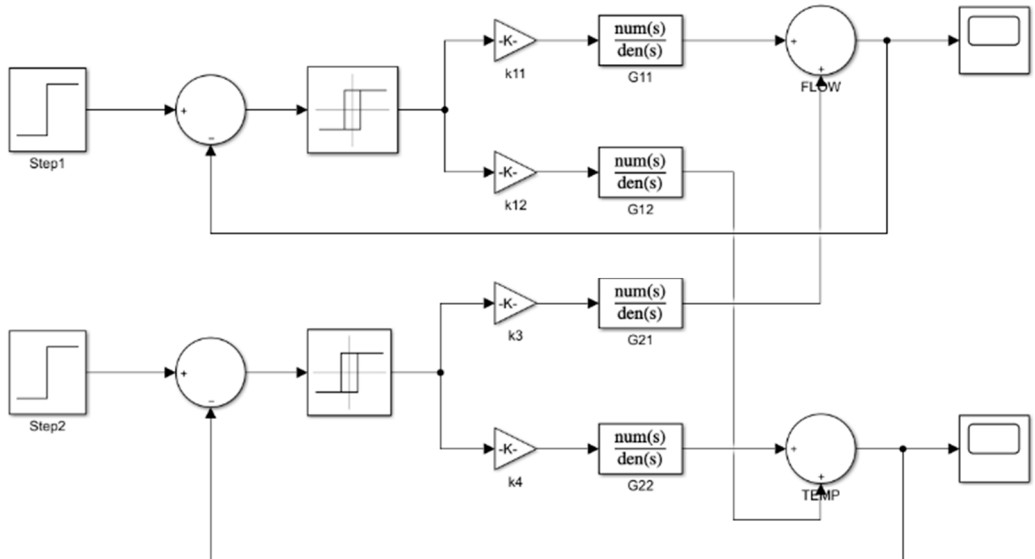

**Figure 4.** Simulation block diagram of environment control system (ECS) with relay feedback.

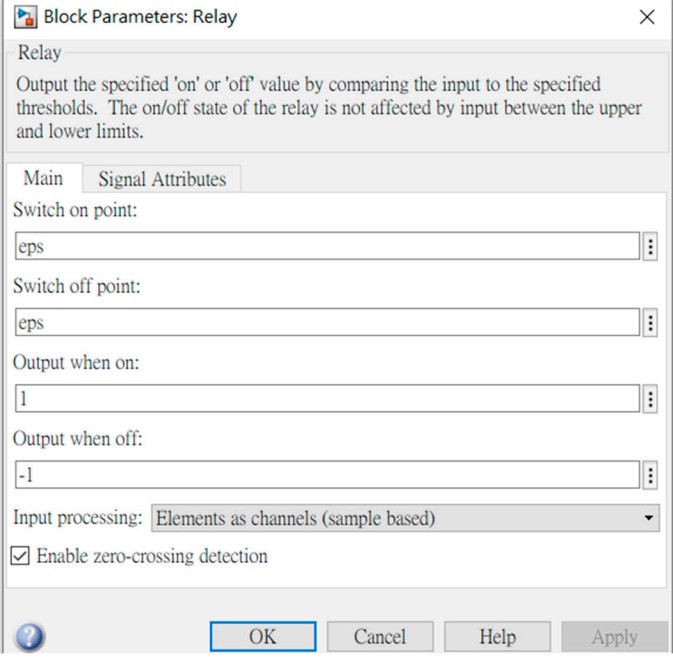

**Figure 5.** Parameter setting for relay block.

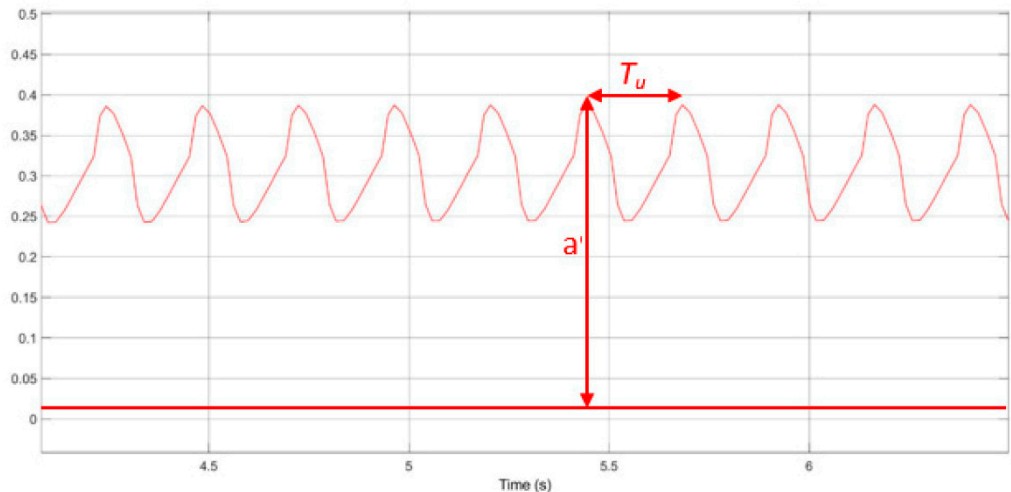

**Figure 6.** Step response of ECS output flow signal with relay feedback method.

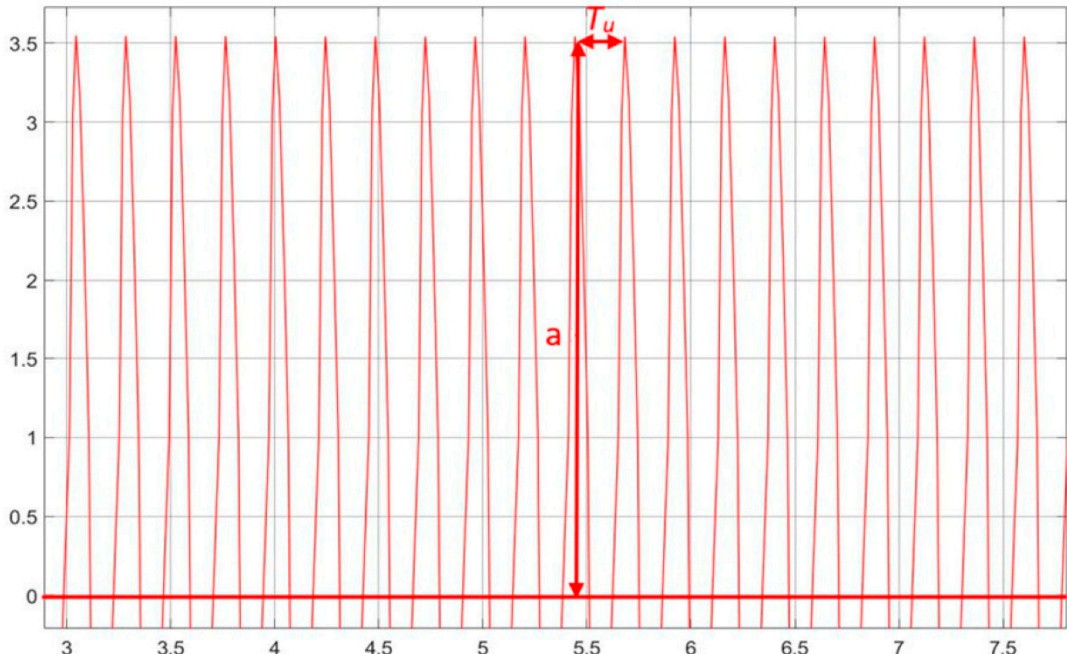

**Figure 7.** Step response of ECS output temperature signal with relay feedback method.

### 3.2. PSO

In the application presented in this study, the parameters suggested by learning factors and used in the algorithm are listed in Table 2. We have two sets of PID gain parameters, including the three gain parameters of the flow and temperature controller in each one. In addition, the definition of optimization is the smallest output error of the controller with two sets of PID gain parameters [35]. The code of PSO fitness function with comments in MATLAB is in Appendix A.

**Table 2.** Parameters of particle swarm optimization (PSO).

| Parameter | Values |
|---|---|
| Acceleration Constant $C_1$ | 1.2 |
| Acceleration Constant $C_2$ | 1.2 |
| Inertia weight factor $w$ | 0.9 |
| Number of particles | 100 |
| Number of iterations | 50 |

### 3.3. SSO

The MATLAB code of the fitness function in SSO is the same as that in PSO. The parameters found by experiences and used in the SSO algorithm are listed in Table 3.

**Table 3.** Parameters of simplified swarm optimization (SSO).

| Parameter | Values |
|---|---|
| $C_w$ | 0.55 |
| $C_p$ | 0.75 |
| $C_g$ | 0.95 |
| $N_{pop}$ | 100 |
| $N_{gen}$ | 100 |

### 3.4. Simulation by Fuzzy Logic Designer Package in MATLAB

Figures 8–12 show detailed setting about the proposed fuzzy logical controller. The fuzzy logic designer package in MATLAB is implemented using the Mamdani fuzzy inference method. We set $K_p$, $K_i$, and $K_d$, respectively, as shown in Figure 8. Each interval setting of membership function E, membership function ΔE, and membership function U are shown in Figures 9–11, which include the ranges, names, types, and parameters of the membership functions of the seven fuzzy sets, as listed in Table 4. There are also three fuzzy controller rule tables for $\Delta K_P(t)$, $\Delta K_i(t)$, and $\Delta K_d(t)$, as shown in Figure 12 and Table 5 [36–38].

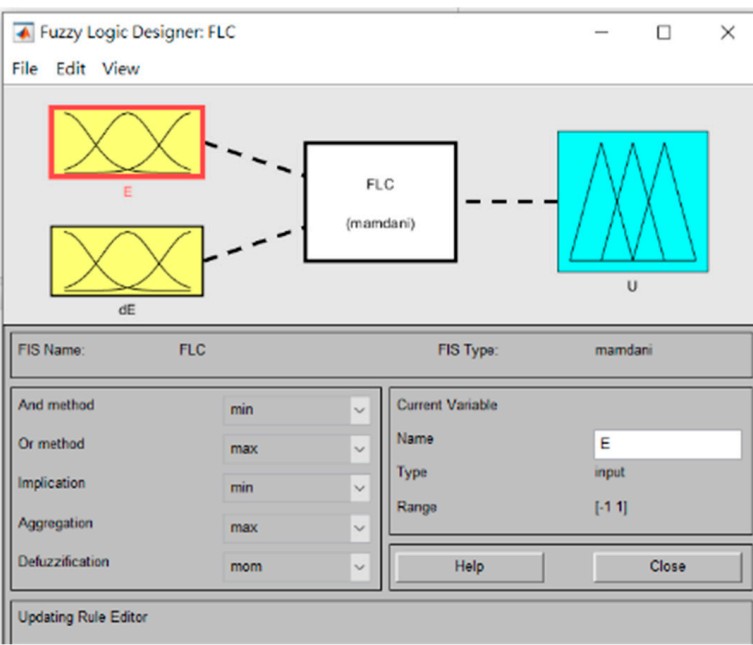

**Figure 8.** Fuzzy inference block.

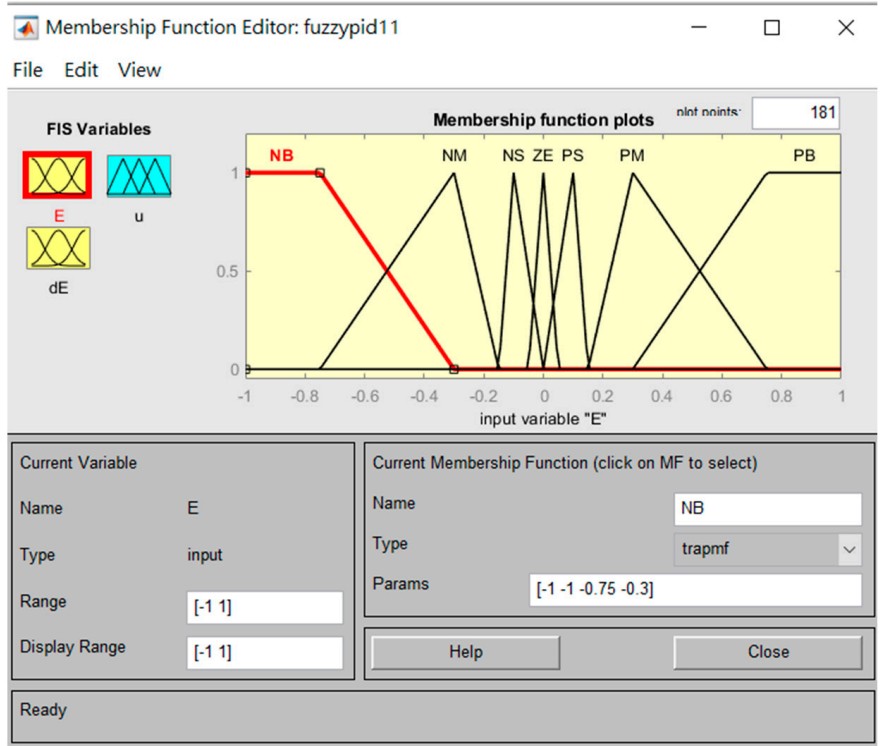

**Figure 9.** Membership function for input variable "E".

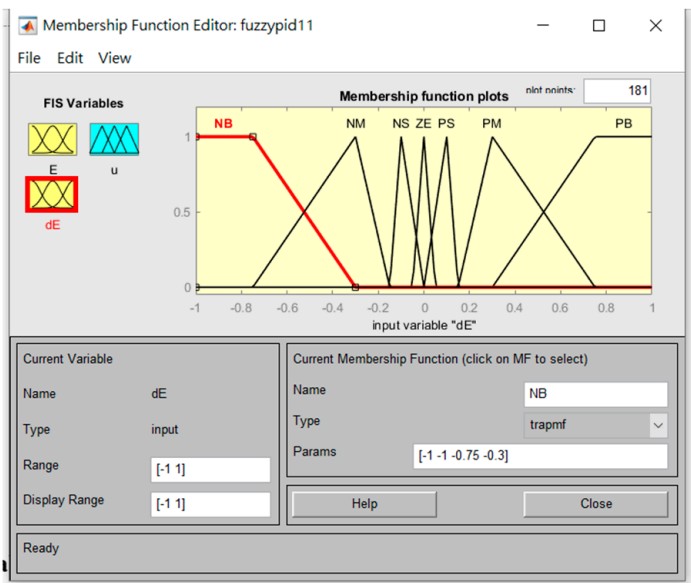

**Figure 10.** Membership function for input variable "ΔE".

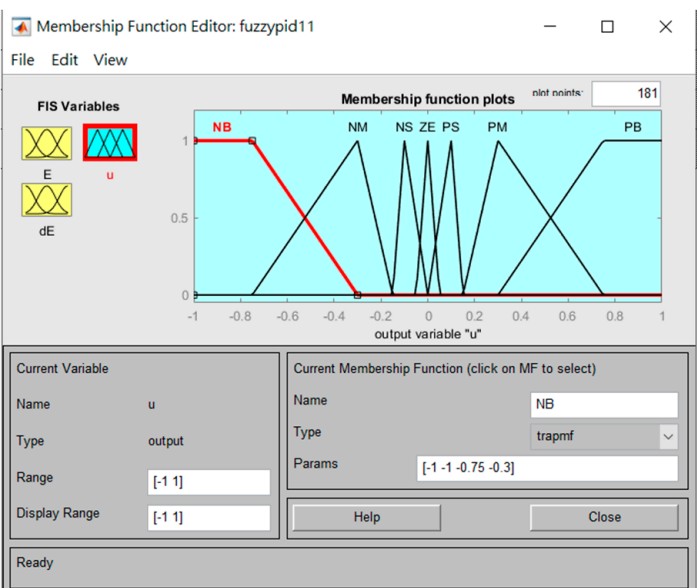

**Figure 11.** Membership function for output variable "U".

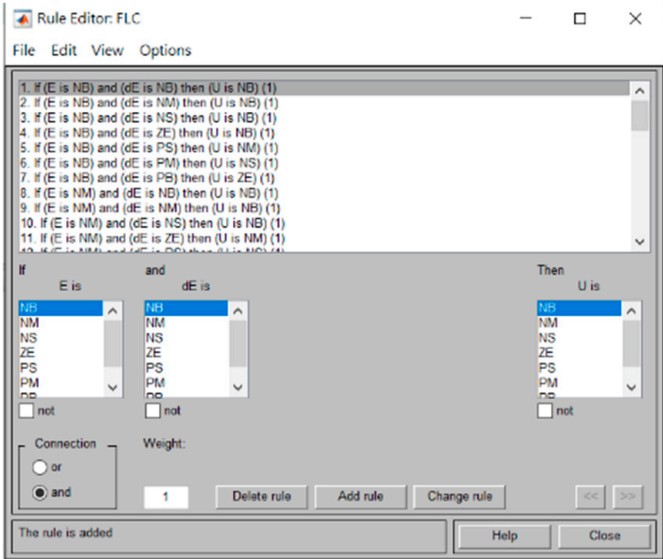

**Figure 12.** Rule table of fuzzy controller (49 rules).

**Table 4.** Table of membership function for input variable "E", "ΔE", and "U".

| Range | | [−1 1] |
|---|---|---|
| **Display Range** | | [−1 1] |
| **Name** | **Type** | **Params** |
| NB | Trapmf | [−1 −1 −0.75 −0.3] |
| NM | Trimf | [−0.75 −0.3 −0.15] |
| NS | Trimf | [−0.15 −0.1 0] |
| ZE | Trimf | [−0.05 0 0.05] |
| PS | Trimf | [0 0.1 0.15] |
| PM | Trimf | [0.15 0.3 0.75] |
| PB | Trapmf | [0.3 0.75 1 1] |

**Table 5.** Fuzzy rule base table of $\Delta K_P(t)$, $\Delta K_i(t)$, $\Delta K_d(t)$.

| E＼dE | NB | NM | NS | ZO | PS | PM | PB |
|---|---|---|---|---|---|---|---|
| NB | PB/NB/PS | PB/NB/NS | PM/NM/NB | PM/NM/NB | PS/NS/NB | ZO/ZO/NB | ZO/ZO/PS |
| NM | PB/NB/PS | PB/NB/NS | PM/NM/NB | PS/NS/NM | PS/NS/NM | ZO/ZO/NS | NS/ZO/ZO |
| NS | PM/NB/ZO | PM/NM/NS | PM/NS/NM | PS/NS/NM | ZO/ZO/NS | NS/PS/NS | NS/PS/ZO |
| ZO | PM/NM/ZO | PM/NM/NS | PS/NS/NS | ZO/ZO/NS | NS/PS/NS | NM/PM/NS | NM/PM/ZO |
| PS | PS/NM/ZO | PS/NS/ZO | ZO/ZO/ZO | NS/NS/ZO | NS/PS/ZO | NM/PM/ZO | NM/PB/ZO |
| PM | PS/ZO/PB | ZO/ZO/NS | NS/PS/PS | NM/PS/PS | NM/PM/PS | NM/PB/PS | NB/PB/PB |
| PB | ZO/ZO/PB | ZO/ZO/PM | NM/PS/PM | NM/PM/PM | NM/PM/PS | NB/PB/PS | NB/PB/PB |

### 3.5. Simulink Simulation

The experimental simulation in this study was completed by MATLAB2018b and Simulink. The model is executed in a personal computer environment with an Intel Core i7, 2.60 GHz, and memory of 12 GB, and its computing time is based on CPU seconds.

#### 3.5.1. Simulink Simulation of Z–N, PSO, and SSO

Figure 13 shows the temperature and flow controller model designed using Simulink for the Z–N, PSO, and SSO algorithms. The three PID gain parameters calculated by the Z–N, PSO, and SSO algorithms are input into the PID controller, as shown in Figure 14. Figure 14 shows conventional PID gain parameters setting for Figure 13 block PID(s). Simulink was then applied to simulate the temperature and flow responses by inputting the unit step response.

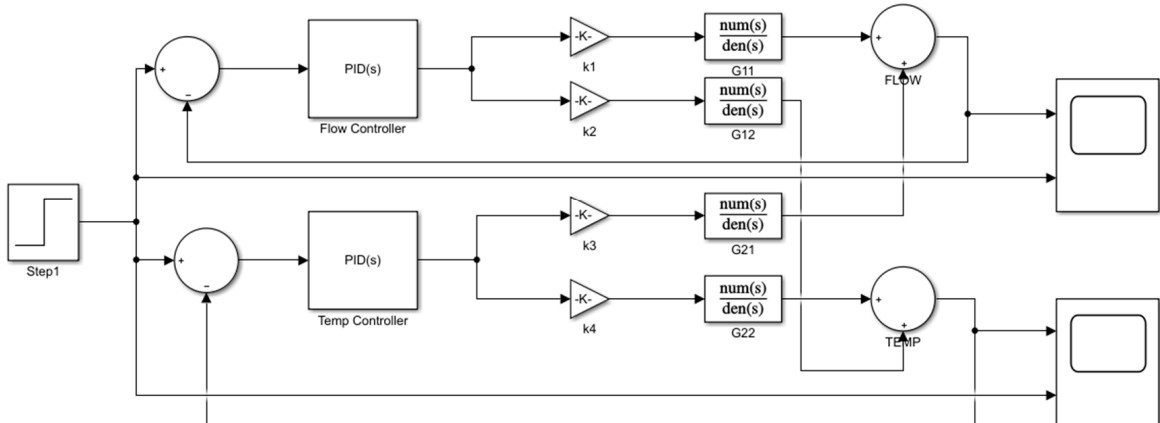

**Figure 13.** Simulink simulation of flow and temperature system (Z–N, PSO, and SSO).

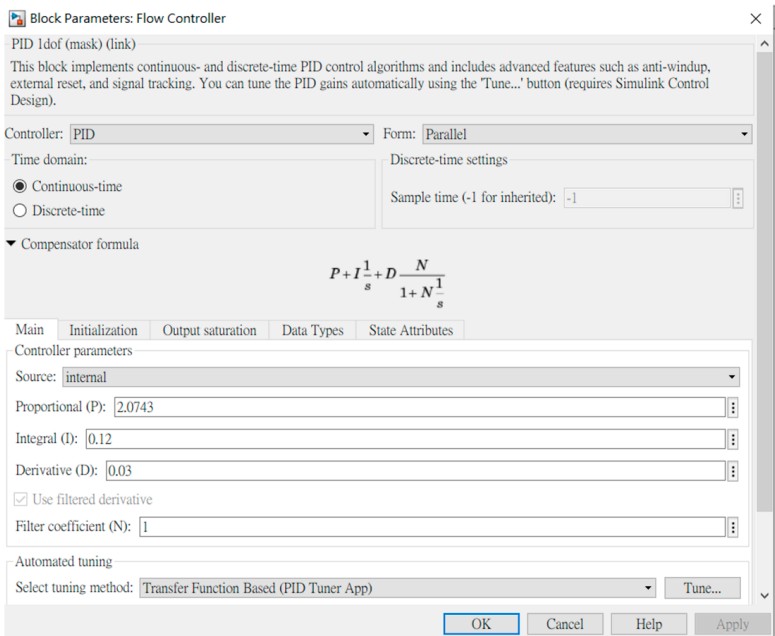

**Figure 14.** Diagram of proportional-integral-derivative (PID) gain parameters setting.

### 3.5.2. Simulink Simulation of Fuzzy PID

Figure 15 shows a Simulink model that utilizes the fuzzy logic designer to tune the initial values of PID gain parameters. The PID gain parameters calculated by the PSO and SSO algorithms are used as the initial values for comparison with those of the fuzzy logic designer combined with the PSO and SSO algorithms. As we have mentioned above, the three gain control parameters in PID control do not necessarily have to be used at the same time. We can only use PI control to reduce computational burden or even get better control, by removing the third fuzzy controller in flow controller and temperature controller.

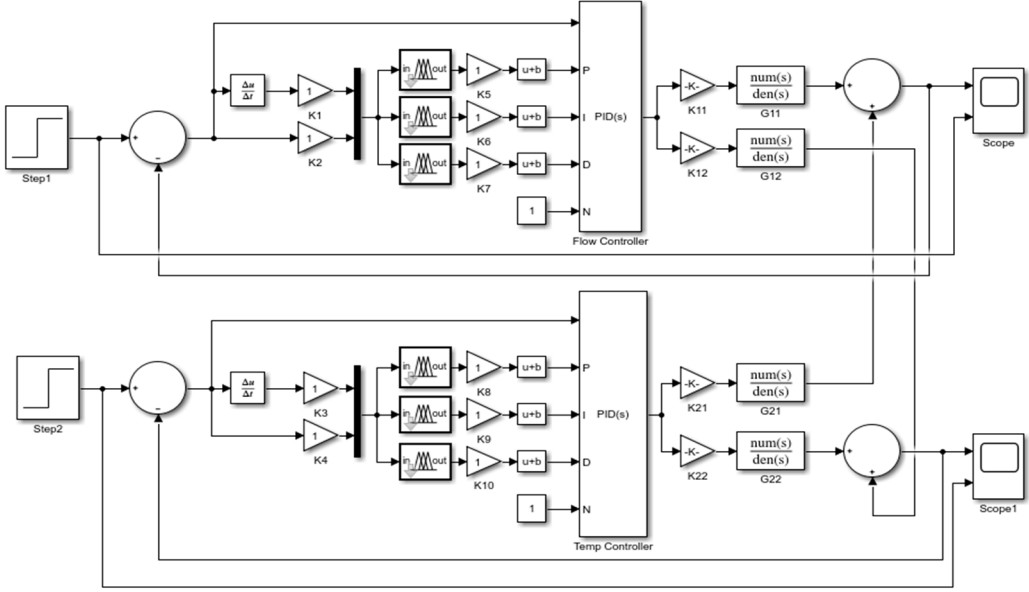

**Figure 15.** Simulink simulation of flow and temperature system (fuzzy PID).

In addition, the ten gain values of K1, K2, K3, K4, K5, K6, K7, K8, K9, and K10 are called scaling factors, which vary by application. Designers decide each variable to reach the results. Their aim

is to set the membership functions between −1 and 1, which strongly limits the oscillation of the controlled system.

## 4. Experimental Results

The following figures depict the responses of flow and temperature after receiving the unit step function input in Simulink using Z–N PID (Figures 16 and 17), PSO PID and SSO PID (Figures 18 and 19), PSO fuzzy PID and SSO fuzzy PID (Figures 20 and 21), and SSO fuzzy PI (Figures 22 and 23). The results are compared in Table 6.

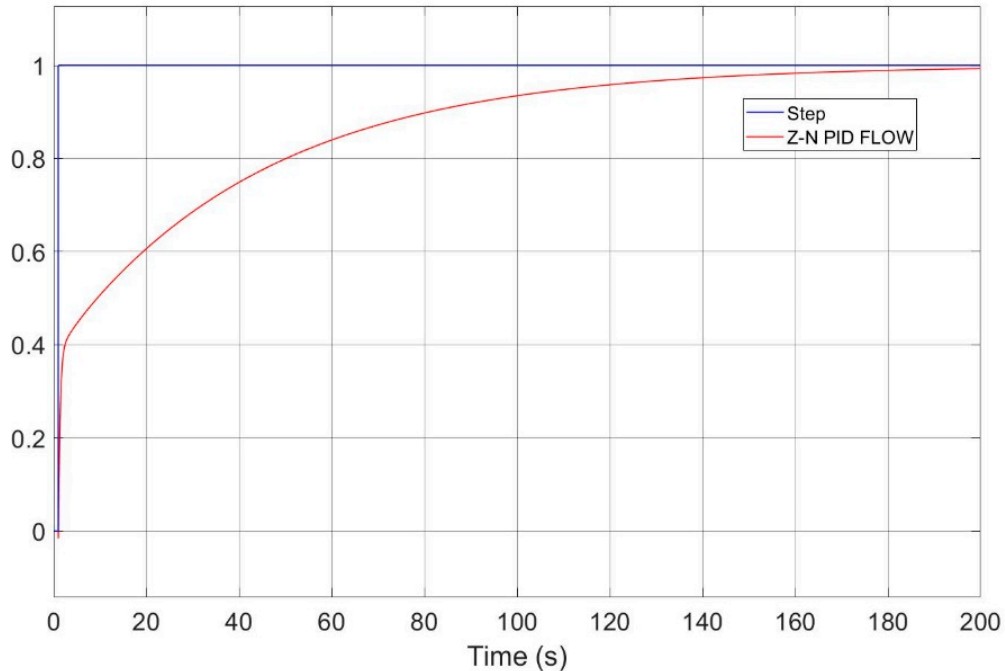

**Figure 16.** Flow versus time plot for PID tuned with Zeigler–Nicholas.

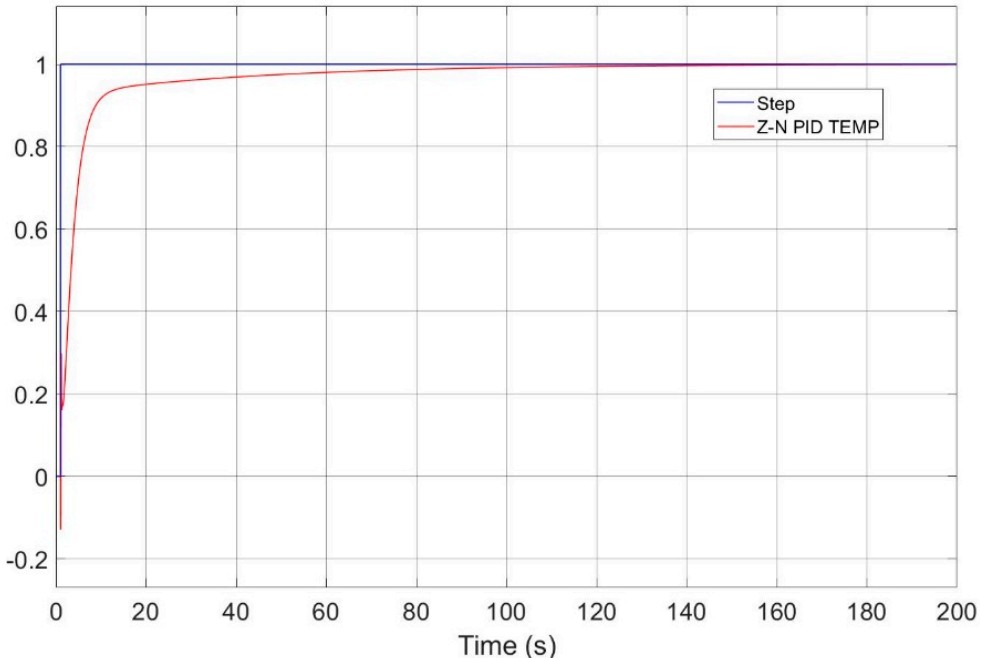

**Figure 17.** Temperature versus time plot for PID tuned with Zeigler–Nicholas.

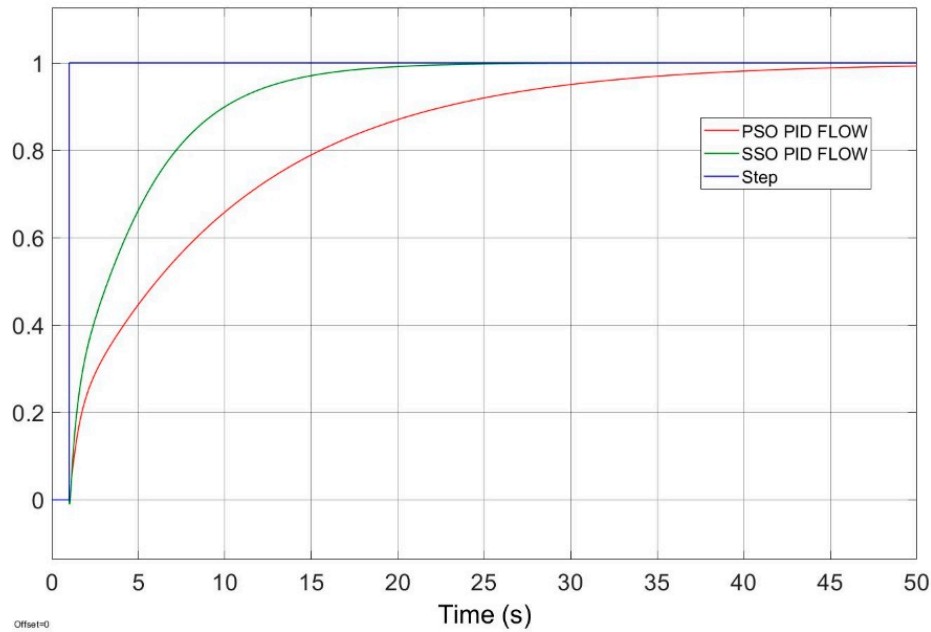

**Figure 18.** Flow versus time plot for PID tuned with PSO and SSO.

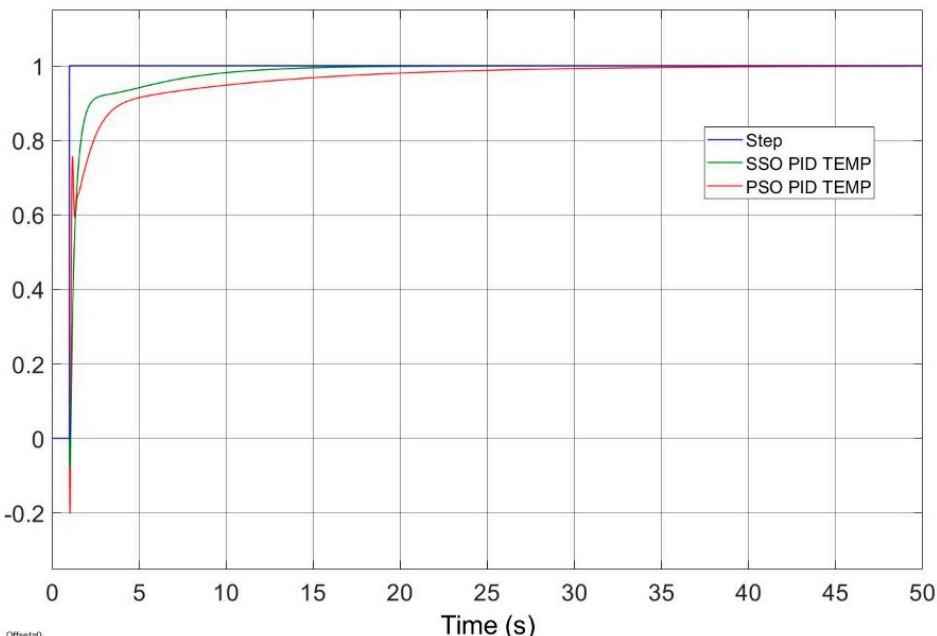

**Figure 19.** Temperature versus time plot for PID tuned with PSO and SSO.

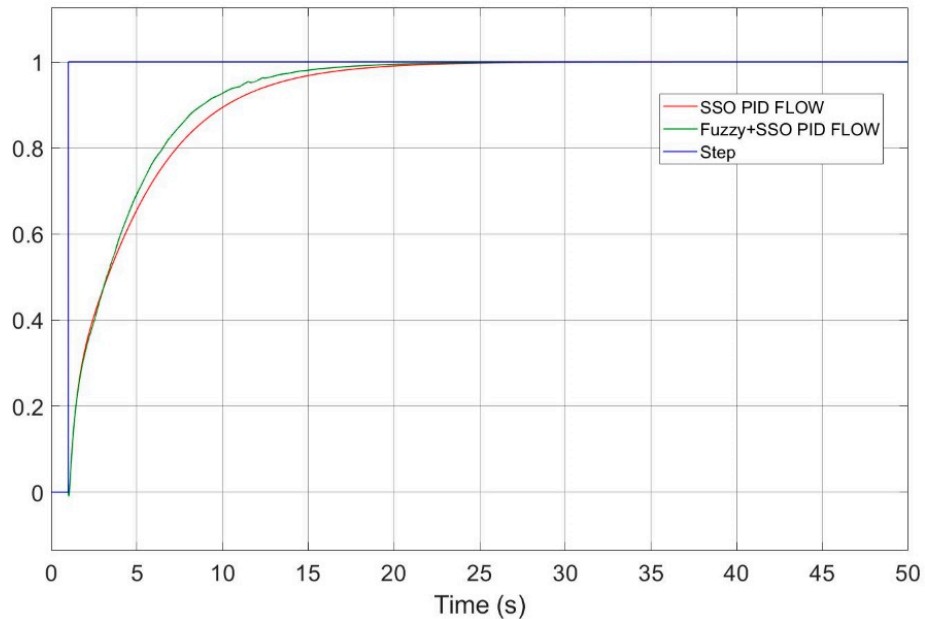

**Figure 20.** Flow versus time plot for PID tuned with SSO and SSO fuzzy.

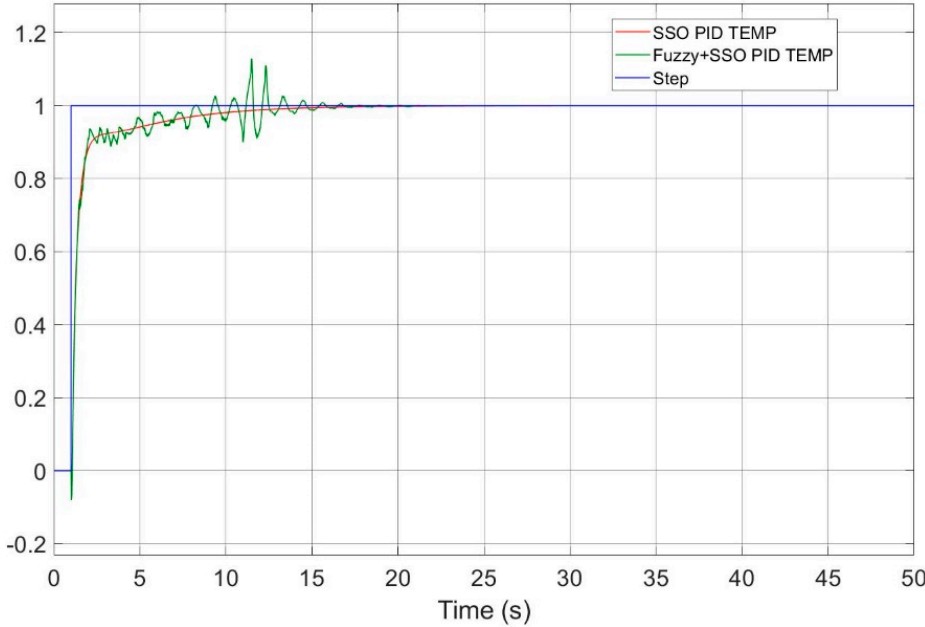

**Figure 21.** Temperature versus time plot for PID tuned with SSO and SSO fuzzy.

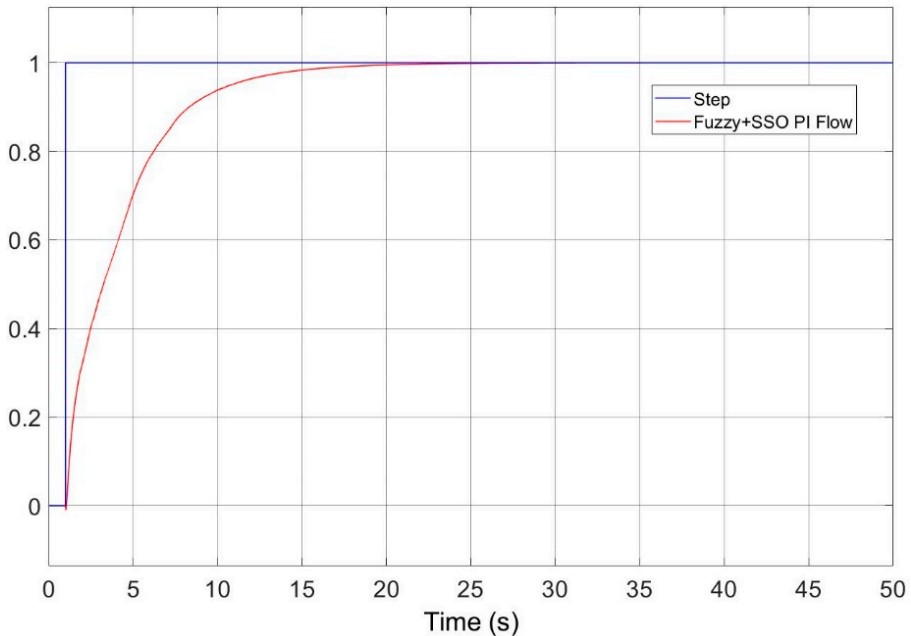

**Figure 22.** Flow versus time plot for PI tuned with SSO and SSO fuzzy.

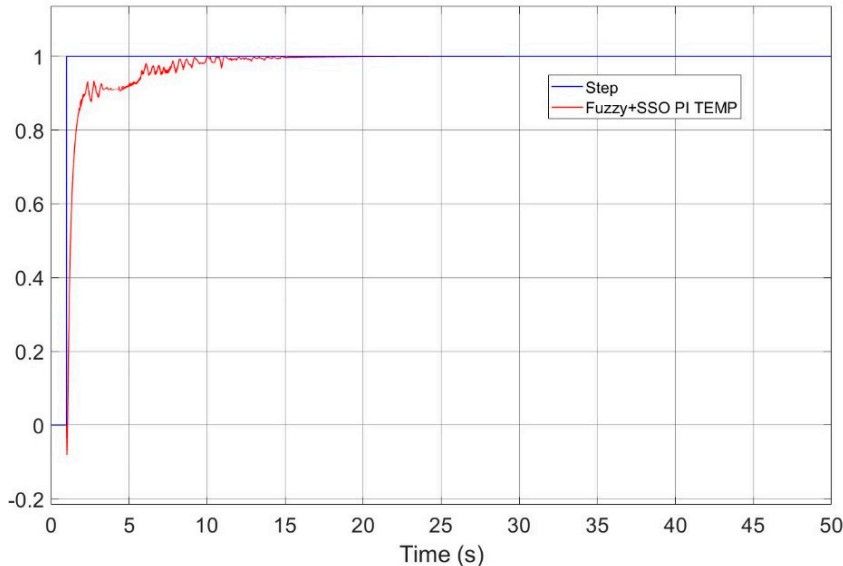

**Figure 23.** Temperature versus time plot for PI tuned with SSO and SSO fuzzy.

**Table 6.** $K_p$, $K_i$, $K_d$, and ITAE of different tuning methods. *ITAE: Integral of absolute error multiplied by time.

| Method | $K_p$ | $K_i$ | $K_d$ | *ITAE (200 s) |
|---|---|---|---|---|
| Z–N PID (flow controller) | 2.0743 | 0.1200 | 0.0300 | 1148 |
| Z–N PID (temp controller) | 0.2158 | 0.1200 | 0.0300 | 150 |
| PSO PID (flow controller) | 0.8536 | 0.3810 | 0.0620 | 95.79 |
| PSO PID (temp controller) | 0.2410 | 0.4168 | 0.0063 | 15.21 |
| SSO PID (flow controller) | 0.9857 | 0.8732 | 0.3767 | 19.42 |
| SSO PID (temp controller) | 0.0419 | 0.7976 | 0.1098 | 3.468 |
| PSO fuzzy PID (flow controller) | 0.8536 | 0.3810 | 0.0620 | 50.09 |
| PSO fuzzy PID (temp controller) | 0.2410 | 0.4168 | 0.0063 | 10.63 |
| SSO fuzzy PID (flow controller) | 0.9857 | 0.8732 | 0.3767 | 15.79 |
| SSO fuzzy PID (temp controller) | 0.0419 | 0.7976 | 0.1098 | 4.23 |
| SSO fuzzy PI (flow controller) | 0.9857 | 0.8732 | – | 14.68 |
| SSO fuzzy PI (temp controller) | 0.0419 | 0.7976 | – | 3.316 |

From the above-mentioned figures and table, we can clearly realize the advantages and disadvantages of the controllers designed by various methods. The performances ordered from high to low correspond to SSO fuzzy PI, SSO fuzzy PID, SSO PID, PSO fuzzy PID, PSO PID, and Z–N PID. From Figures 16 and 17, it can be seen that the response time for Z–N PID to converge to 1 is the slowest, nearly 200 s, and thus, it is not suitable for practical applications where the response time is emphasized. The experimental results of PSO PID are not always excellent in the simulation process and are easily limited to the local optimum. Moreover, its run time is longer than that of SSO, but obviously, the time required to converge to 1 has dropped to approximately 50 s, as shown in Figures 18–21.

SSO PID performs not only with faster run time, but also with a higher probability of obtaining an excellent solution. Its settling time is shorter than that of PSO, as shown in Figures 18 and 19. By combining the fuzzy theory with PSO and SSO, it was found that the experimental results are both more optimized than the original ones as can be seen from Figures 20–23 and the values of the integral of absolute error multiplied by time (ITAE) in Table 6. PSO fuzzy PID, SSO fuzzy PID, and SSO fuzzy PI not only inherit the advantages of PSO PID and SSO PID, but also change dynamically in real time. The PID gain parameters are adjusted by the proposed strategies, and hence, the settling time is shortened successfully. Moreover, SSO fuzzy PI performs better than SSO fuzzy PID in run time and ITAE value, as shown in Figures 22 and 23 and Table 6. Therefore, SSO fuzzy PI stands out among all strategies.

## 5. Conclusions

This study designs and compares the PID controllers in MATLAB and Simulink by using five methods, including Ziegler–Nicolas PID tuning, PSO PID, SSO PID, and the combination of the fuzzy theory with PSO PID and SSO PID. It is noticeable that this study pioneeringly implements SSO in a fuzzy PID controller. To demonstrate the capabilities of both system response speed and better performance with minimum errors, ITAE, which is the integral of absolute error multiplied by time, is utilized as our fitness function. We successfully applied the fitness functions in the PSO and SSO algorithms to resolve the interactive effect of the MIMO system. All the gain parameters of the two controllers are considered simultaneously, instead of a single flow controller or temperature controller. Therefore, our designed control system can take into account the benefits of both goals at the same time and achieve the optimization results.

From the experimental simulation results in this study, we understood that the fuzzy theory makes it possible to adjust the PID gain parameters dynamically over time and makes the system more robust. In addition to reducing the settling time, the ITAE is also significantly reduced, but the run time becomes longer. This is a shortcoming of the fuzzy theory. However, in practice, advanced hardware equipment may compensate for this problem. In other applications, we refer that the use of the fuzzy theory in control strategies can also provide better domination of nonlinear or time varying control systems. In summary, it has been proved that the SSO algorithm is superior to the PSO algorithm. With the SSO algorithm, the run time, control effects (i.e., ITAE and settling time), and proportion of better solutions are all greater than those with the PSO algorithm.

**Author Contributions:** T.-Y.W. and W.-C.Y. conceived and design the experiments; T.-Y.W. performed the experiments; T.-Y.W. analyzed the data; T.-Y.W., Y.-Z.S. and Y.-Z.J. wrote the paper. All authors have read and agreed to the published version of the manuscript.

**Funding:** This research is supported by Natural Science Foundation of China (61702118).

**Conflicts of Interest:** The authors declare no conflict of interest.

## Appendix A

Function F = fitness(x).
Input kp ki kd of the flow controller

Input KP KI KD of the temperature controller

Let ai bi ci di ei be the parameters of the transfer functions sysi, for i = 1, 2, 3, 4.

sys1 = tf([Kd*d1  Kp*d1+Kd*e1  Kp*e1+Ki*d1  Ki*e1], [Kd*d1+a1  Kp*d1+Kd*e1+b1 Kp*e1+Ki*d1+c1 Ki*e1]); %G11

sys2 = tf([Kd*d2  Kp*d2+Kd*e2  Kp*e2+Ki*d2  Ki*e2], [Kd*d2+a2  Kp*d2+Kd*e2+b2 Kp*e2+Ki*d2+c2 Ki*e2]); %G12

sys3 = tf([KD*d3  KP*d3+KD*e3  KP*e3+KI*d3  KI*e3], [KD*d3+a3  KP*d3+KD*e3+b3 KP*e3+KI*d3+c3 KI*e3]); %G21

sys4 = tf([KD*d4  KD*d4+KD*e4  KP*e4+KI*d4  KI*e4], [KD*d4+a4  KP*d4+KD*e4+b4 KP*e4+KI*d4+c4 KI*e4]); %G22

t = (0:0.01:100)';

S1 = step(sys1,t).

S2 = step(sys2,t).

S3 = step(sys3,t).

S4 = step(sys4,t);

S_flow = S1 + S3;

S_temp = S2 + S4;

err_flow = abs(S_flow-1);

err_temp = abs(S_temp-1).

flow = 0;

temp = 0;

for i = 1:size(err_flow,1)

　flow =flow+err_flow(i)

end

for i = 1:size(err_temp,1)

　temp = temp + err_temp(i).

end

F = flow + temp;

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
