# Peer review of "Using Simplified Swarm Optimization on Multiloop Fuzzy PID Controller Tuning Design for Flow and Temperature Control System"

_applsci, doi:10.3390/app10238472_

Round 1

Reviewer 1 Report

No comment.... The resubmitted manuscrip is suitability for
publication.....

Author Response

Thank you very much for the review.

Reviewer 2 Report

This study proposes the flow and temperature controllers of a cockpit environment 15 control system (ECS) by implementing an optimal simplified swarm optimization (SSO) fuzzy 16 proportional-integral-derivative (PID) control. Tje PID controller dates back to 1910 and Zadeh's fuzzy logic to 1965. Swarm optimization dates to 1995. The main contribution of this study is the implementation of SSO in a fuzzy PI/PID controller.The graphic presentations certainly support a successful study. But, SSO is an EP technique dating from 1995. I would like to see more discussion of the SSO algorithm vs. the PSO algorithm for particle swarms to be sure that the Matlab equation fitting is not just overfitting on the part of Matlab. The authors have not addressed this possibility.

Author Response

This manuscript is a resubmission of an earlier submission. The following is a list of the peer review reports and author responses from that submission.

Round 1

Reviewer 1 Report

  • adjust the numbering of equations to one plane,
  • fig. 4 - unreadable text (enlarge),
  • fig. 6 - unreadable text y-axis,
  • fig. 7 - unreadable text x and y-axis,
  • fig. 7 - repair double y-axis,
  • fig. 13 and 15 - unreadable text,
  • move text on line 451 to p. 18 (to the table)
  • In Table 6 in the last two rows only from the "Kd" column it is visibler that it is the PI controller and in the previous 2 rows is the PID controller.
    Please add to the description (in the "Methods" column) of PI F SSO and previously 2 rows of PID F SSO.

    - suggestions for Authors:
    1. From the article it can be seen that the authors did not achieve much better results if they use fuzzy SSO and SSO itself.
    2. Why didn't the authors suggest a PI controller (and also a fuzzy PI controller) that, in our experience, could achieve better results?
    3. If the authors want to stay with the PID controller, then let the authors also design the PI controller and compare the results with the fuzzy controller (so check whether use the PI or the PID controller).
    4. It would be appropriate to compare the achieved results with other published optimization methods (other authors).

Reviewer 2 Report

The authors proposed the flow and temperature controllers of a cockpit environment control system (ECS) by implementing an optimal simplified swarm optimization (SSO) fuzzy proportional-integral-derivative (PID) control, and the proposed controller verified by comparing thoes by implementing some other methods. However I have many the following questions for this study.

1) I have doubts about the effectiveness of the results of this paper. As far as I know, real-time control based on a heuristic algorithm takes a lot of time to find a solution, so it raises questions about its effectiveness in using it in real applications. Moreover, in this paper, the authors were said that the existing heuristic-based algorithms have the limitation of using local solutions due to time limitations, so they have solved this problem by proposed their method. I wonder if it really can be improved. I would like to hear from the authors on this.

2) In terms of control, the purpose of this paper is unclear. It is unclear whether the goal is to improve the response speed in transient response or to minimize the errors in steady-state response. The authors were concluded that the response speed of the controller was improved, but the objective function of optimization adopts an objective function that minimizes errors. It seems like a study whose purpose is unclear. There is a need for a clear explanation of the authors.

3) The equations are not clear, and the equation development is not smooth. This is a problem for expression of expressions that do not take into account the convenience of the reader, e.g., the size of the letter is uneven or the meaning is not properly expressed. In addition, the bigger problems are that there are many parts that do not fit their meanings. For example, Equation (3) and Equation (4) cannot be the same. Because Equation (3) is an equation in the time domain, and Equation (4) is an equation in the frequency domain. Additional parts are needed to express that the two equations are the same. This is just one example. Overall, it is necessary to correct and confirm the equations.

4) Overall, the size of the characters in the figures is too small to read. I think to need improvement.

5) The code on page 10 is questionable what information was put in to represent. If you wanted to express a Fitness function, why do you write code? Just putting in a formula will make it easier for the reader to understand and see.

6) It is questionable what Figures 5, 8-12, and 14 wants to show.

7) There is no explanation for why parameters of PSO and SSO algorithms were selected as shown in Tables 2 and 3. Are the optimal parameters selected? It's not convincing. I think to need the parameter study.

8) The analysis of the results of the comparative experiment is insufficient. In this paper, the comparative experiments are very important and essential. Nevertheless, the analysis to verify the propose method is little.

9) In figures of the results of the experiment, it is difficult to identify the partial plot corresponding to the transient response. I think to need the enlarged figures.

10) There is a lack of analysis of numerical data on the results of the experiment, and there are no tables to show these.

11) This paper verifies the performance of controller based on heuristic algorithm through experiment. Heuristic algorithms are mostly based on probabilities. They can't get the same result every time. Therefore, statistical data presentation and analysis are necessary. However, this study does not show.

12) There is a lack of references to the latest research papers. In addition, there is no comparative study with the latest researches.

Reviewer 3 Report

This study tried to design and compare the PID controllers in MATLAB and Simulink by using five methods.  They are Ziegler–Nicolas PID tuning, particle swarm optimization (PSO) PID, SSO PID, and the combination of the fuzzy theory with PSO PID and SSO PID, respectively. In addition, the authors mentioned they proposes the flow and temperature controllers of a cockpit environment control system (ECS) by implementing an optimal simplified SSO fuzzy PID control.  Moreover, they indicated that the ECS model is considered as a multiple-input multiple-output (MIMO) and second-order dynamic system, which is interactive.  How the authors demonstrated those interactions will not influence the control? 

How to approve the main contribution of this study is the pioneering implementation of SSO in a fuzzy PI/PID controller should be addressed. 

Moreover, the reason of adding the original gain parameters Kp, Ki, and Kd in PID controller with delta values should be addressed in details.  How to approve and demonstrate this can make your control system more accurate, adaptive, and robust.

The relevant literatures should be reviewed, discussed and summarized. 

Round 2

Reviewer 3 Report

Extensive editing of English language and  style are required. 
